# In silico studies provide new structural insights into trans-dimerization of $\beta_1$ and $\beta_2$ subunits of the Na+, K+-ATPase

Gema Ramírez-Salinas[1], Liora Shoshani[2]*, Jorge L. Rosas-Trigueros[3], Christian Sosa Huerta[2], Marlet Martínez-Archundia[1]*

**1** Laboratorio de Diseño y Desarrollo de Nuevos Fármacos e Innovación Biotecnológica (Laboratory for the Design and Development of New Drugs and Biotechnological Innovation), Sección de Estudios de Posgrado e Investigación, Escuela Superior de Medicina, Instituto Politécnico Nacional, Ciudad de México, México, **2** Department of Physiology, Biophysics, and Neurosciences, Center for Research and Advanced Studies (Cinvestav), Mexico City, Mexico, **3** Laboratorio Transdisciplinario de Investigación en Sistemas Evolutivos, ESCOM, Instituto Politécnico Nacional, Mexico City, Mexico

* shoshani@fisio.cinvestav.mx (LS); mtmartineza@ipn.mx (MMA)

**Data availability statement:** All relevant data are within the manuscript and its Supporting information files.

## Abstract

The Na+, K+-ATPase is an electrogenic transmembrane pump located in the plasma membrane of all animal cells. It is a dimeric protein composed of $\alpha$ and $\beta$ subunits and has a third regulatory subunit ($\gamma$) belonging to the FXYD family. This pump plays a key role in maintaining low concentration of sodium and high concentration of potassium intracellularly. The $\alpha$ subunit is the catalytic one while the $\beta$ subunit is important for the occlusion of the K+ ions and plays an essential role in trafficking of the functional $\alpha\beta$ complex of Na+, K+-ATPase to the plasma membrane. Interestingly, the $\beta_1$ and $\beta_2$ (AMOG) isoforms of the $\beta$ subunit, function as cell adhesion molecules in epithelial cells and astrocytes, respectively. Early experiments suggested a heterotypic adhesion for the $\beta_2$. Recently, we reported a homotypic trans-interaction between $\beta_2$-subunits expressed in CHO cells. In this work we use In Silico methods to analyze the physicochemical properties of the putative homophilic trans-dimer of $\beta_2$ subunits and provide insights about the trans-dimerization interface stability. Our structural analysis predicts a molecular recognition mechanism of a trans-dimeric $\beta_2 - \beta_2$ subunit and permits designing experiments that will shed light upon possible homophilic interactions of $\beta_2$ subunits in the nervous system.

## Introduction

The Na+, K+-ATPase, a ubiquitous plasma-membrane ion pump plays a crucial physiological role in all animal cells. Indeed, the resultant ion and electrochemical gradients are essential for many physiological processes and in the brain, about 50% of the ATP is consumed by the Na+, K+-ATPase [1]. Na+, K+-ATPase is a P-type ATPase, an oligomeric enzyme that consists of three subunits: $\alpha$, $\beta$ and $\gamma$ [2,3]. This work is focused on the $\beta$-subunit. The Na+, K+-ATPase $\beta$ subunit is part of the functional core of the pump and is required for its trafficking to the plasma membrane. Mammals express three $\beta$ subunit isoforms, $\beta_1$, $\beta_2$ and $\beta_3$. It has a small intracellular, N-terminal domain (30 amino acids), a single transmembrane

**Funding:** CONAHCYT (Proyecto Ciencia Frontera CF-2023-G-1454). The funders had no role in study design, data collection and analysis, decision to publish, or preparation of the manuscript.

**Competing interests:** The authors have declared that no competing interests exist.

helix, and a large extracellular, C-terminal domain of about 240 amino acids [4,5]. The different $\beta$ isoforms have distinct tissue and cell-type specific expression profiles [6,7]. There are three conserved disulfide bonds in the extracellular domain, which are important for forming a stable pump [8], and the extracellular domain has three, eight, and two glycosylation sites in $\beta_1$, $\beta_2$, and $\beta_3$, respectively [9,10]. Functionally, $\beta_2$ has the strongest effects on the kinetic properties of the pump, reducing the apparent potassium affinity and raising the extracellular sodium affinity compared to $\beta_1$ and $\beta_3$ [11]. The different $\beta$ isoforms and the variation in their post-translational modifications facilitate regulated Na$^+$, K$^+$-ATPase activity, adapted to different tissues and to environmental changes. The $\beta$ subunit is important for the occlusion of the K+ ions and plays an essential role in trafficking of the functional $\alpha\beta$ complex of Na$^+$, K$^+$-ATPase to the plasma membrane [12]. Apart from the role of $\beta$ subunit in regulating the pump activity, a role in cell-cell adhesion has been also proposed [13]. With this regards, [13] have suggested that the Na$^+$, K$^+$-ATPase acts as a cell adhesion molecule by binding to the Na$^+$, K$^+$-ATPase molecule of a neighboring cell by means of trans-dimerization of their $\beta_1$ subunits. Following, it was demonstrated that a direct homotypic interaction between $\beta_1$-subunits of neighboring cells, takes place between polarized epithelial cells [14,15] identified the amino acid region crucial for the species-specificity of this trans-interaction [16] completed the description of the adhesion interface between the extracellular-domains of the dog $\beta_1$-subunits. Earlier, the group of Schachner identified an adhesion molecule on glia (AMOG) that functions as a neural recognition molecule mediating neuron-glia interactions that promotes migration and neurite outgrowth [17,18]. This adhesion molecule was later identified as the $\beta_2$-subunit of the Na$^+$, K$^+$-ATPase and was named $\beta_2$/AMOG [19]. Their works suggested a heterophilic interaction between AMOG and an unknown molecule at the neuron membrane [19,20]. The crystal structure analysis of the Na$^+$, K$^+$-ATPase $\beta_1$ subunit in the E2 state as published by Shinoda and colleagues, marked a significant milestone by revealing the atomic structure of the extracellular domain of the $\beta_1$ subunit (PDB: 2ZXE) [21]. Notably, the extracellular C-terminal domain of the protein adopts an Ig-like $\beta$-sheet sandwich configuration, consistent with in silico predictions [22]. Intriguingly, although many adhesion and non-adhesion proteins feature domains with an immunoglobulin-like (Ig-like) topology, structural alignments of the $\beta_1$-subunit extracellular domain against well-studied cell adhesion molecules do not reveal any structural homologs to $\beta$ subunits. Upon detailed examination, three distinctive features of the $\beta$ subunit family members emerge: 1. The Ig-like fold with a unique topology, interrupted by a long $\alpha$-helix secondary structure. 2. An atypical $\beta$-sheet disposition in relation to classical Ig folds. 3. The $\beta$ subunit fold contains extensive loops, resulting in a length twice that of a typical Ig domain. Furthermore, the structural relationship between the $\beta_1$ subunit and the catalytic $\alpha$ subunit suggests that the C-terminal fold must exhibit greater rigidity compared to the typical flexibility seen in adhesion domains, such as cadherin-domains [23]. Further works including mutational analysis combined with In Silico studies have identified the residues at the dog $\beta_1$ surface that participate in $\beta_1 - \beta_1$ interaction [15,16]. Although It is well accepted that both isoforms $\beta_1$ and $\beta_2$ function as adhesion molecules in epithelia and in the nervous system, respectively there is almost no information regarding the adhesion mechanism of $\beta_2$ /AMOG isoform. Very recently it has been published that $\beta_2$ acts as an homophilic adhesion molecule when expressed in CHO fibroblasts and MDCK epithelial cells [24]. Cell-cell aggregation, protein-protein interaction assays as well as In Silico studies were carried out to confirm cell-cell adhesion mediated by $\beta_2 - \beta_2$ trans-interaction. With these results the authors localized the putative interacting surface in a docked model and suggested that the glycosylated extracellular domain of $\beta_2$/AMOG, can make an energetically stable trans-interacting homodimer. In the present

work we have built homotypic dimers of the human $\beta_1$ and $\beta_2$ subunits by employing protein-protein docking analysis, and submitted them to molecular dynamics simulations (MDS) which provide detailed information about their dimeric conformation and specific differences in their interfaces. We also investigated the role of the glycosylation in the interface stabilization of the human $\beta_1$ and $\beta_2$ dimeric complexes.

## Materials and methods

### Molecular modeling of the monomers of Na$^+$, K$^+$-ATPase $\beta$ subunits in humans

Three dimensional (3D) structures of the $\beta$-subunits of Na$^+$, K$^+$-ATPase: ATP1B1 and ATP1B2 were obtained by employing the Swiss Model Program [25]. For the In Silico studies of both proteins we considered only the extracellular domain of the Na$^+$, K$^+$-ATPase $\beta_1$ and $\beta_2$ subunit.

The glycosylation and disulphide bridges sites on both ATP1B1 and ATP1B2 were taken from the Uniprot database ATP1B1 (P05026) and ATP1B2 (P14415) [26]. For the first protein ($\beta_1$), the following glycosylation sites were considered: N158, N193 and N265, and the disulphide bridges: S126–S149, S159–S175 and S213–S276. For the second protein ($\beta_2$), the following glycosylations were considered: N96, N118, N153, N159, N193, N197 and N238, whereas the following disulphide bridges were considered: S129–S150, S160–S177 and S200–S261. Glycosylations (GlcNAc) and disulphide bridges of each of the proteins were included by means of the CHARMM-GUI Program [27].

### Building the dimers of ATP1B1 and ATP1B2 and validation of the 3D models

Once the monomers were correctly built, the molecular docking of both $\beta_1 - \beta_1$ and $\beta_2 - \beta_2$ subunits was performed by using HDOCK server in order to obtain dimer complexes of each of the proteins. HDOCK predicts the interaction of protein-ligand complexes through hybrid algorithm strategy of template-based and template-free docking [28].

After performing protein-protein docking procedure, dimeric complexes were selected according to the criteria a) most energetically favorable $\beta_1 - \beta_1$ (Docking score -193.04 kcal/mol) and $\beta_2 - \beta_2$ (Docking score -274.99 kcal/mol), by means of the HDOCK Server [28], b) trans orientation in the dimeric complexes.

### Molecular dynamics simulations of dimeric complexes of $\beta_1 - \beta_1$ and $\beta_2 - \beta_2$

MD simulations of both dimers were carried out using CHARMM-GUI Server and considering the commands from the Solution Builder implemented in the mentioned Program. Dimers were in a rectangular waterbox size of 10Å edge distance. A NaCl solution (0.15 M) was integrated in the system by using "Distance" as Ion Placing Method [29]. Periodic Boundary Conditions were implemented as Generating grid information for PME FFT automatically. Equilibration of the systems was done using an NVT ensemble and dynamics input was generated as an NPT ensemble (310 K). MD simulations were run for about 200 ns.

### Analysis of the interfaces of the dimeric complexes of $\beta_1$ and $\beta_2$

Molecular interactions were analyzed in the different protein conformations which include: hydrogen bonds (kJ/MOL), electrostatic energy (kJ/MOL), Van der Waals (kJ/MOL), and Total stabilizing energy (kJ/MOL). All these parameters were calculated through PPCHECK

Software [30] which is a specialized web server useful to identify non-covalent interactions at the interface of protein-protein complexes. Moreover, the percentage of residues in the interface, for both chains (chain A and chain B) was calculated using the Program PDBePISA from the Protein Data Bank in Europe [31].

For this analysis we compared three different conformations at 0 ns, 20 ns, 60 ns, 100 ns, 120 ns and 160 ns. Protein-protein interactions in the interfaces were calculated through PDBsum (http://www.ebi.ac.uk/thornton-srv/databases/pdbsum/) in which interface areas are computed using Program called NACCESS http://wolf.bms.umist.ac.uk/naccess, which is implemented in the Software.

## Prediction of binding free energy through Molecular Mechanics Poisson–Boltzmann Surface Area (MM-PBSA) method

Binding free energies of the dimeric complexes of ATP1B1 and ATP1B2 were calculated by means of the pipeline tool named Calculation of Free Energy (CaFE) which is a useful tool to predict binding affinity of some complexes by using end-point free energy methods [32] with the aim to conduct MM-PBSA calculation [33]. In the MM-PBSA analysis, three main energetic components are calculated. Firstly, the gas-phase energy difference between the complex and the receptor separated. Afterwards, the difference of solvent-accessible surface area (SASA) is measured and the non-polar solvation free energy is calculated. Finally the binding free energy is added throughout an ensemble conformations. By means of this analysis we were able to get an insight about non-bonded interactions such as Van der Waals, electrostatic, among other parameters.

## Principal component analysis (PCA)

PCA has become a popular method to reduce the dimensionality of a complex system and has been previously applied to G protein-coupled receptors (GPCRs) [34]. This method diagonalizes the two-point covariance matrix, thus removing the instantaneous linear correlations among the atomic positions. It has been shown that a large part of the system's fluctuations can be described in terms of only a few of the eigenvectors obtained, usually those corresponding to the largest eigenvalues. The principal components are the product of these eigenvectors with the mass weighted coordinates of the molecule and can be used as reaction coordinates and to obtain free energy surfaces of the system, among other analysis that can be performed of this representation of the conformational behavior. We used the dihedral angle principal component analysis (dPCA) version, as modifications in dihedral angles lead often to more dramatic conformational changes than movements in atomic cartesian coordinates [35]. The calculations were performed using the Carma program [36].

## Free energy landscapes

The calculated dPCA were used to represent the free energy surface of the system, restricting the surface to two dimensions (thus using the first two principal components V1 and V2):

$$\Delta G(V_1, V_2) = -kBT\left[\ln \rho(V_1, V_2) - \ln \rho_{max}\right] \tag{1}$$

where $\rho$ is an estimate of the probability density function obtained from a histogram of the data. $\rho_{max}$ denotes the maximum of the density, therefore $\Delta G = 0$ for the region with the highest density [35].

## PCA-based cluster analysis

The calculated dPCA values for each trajectory frame are used to populate a grid to describe the distribution of these values: the higher the value at a grid point, the larger the number of frames with dPCA values closest to this grid point. Isolated maxima in this distribution map correspond to heavily populated clusters. Cluster number 1 is the cluster with the highest density, which would correspond with the region where $\Delta G = 0$ in the free energy landscape described above.

## Contribution of movements in each of the residues along the trajectories

In the dPCA, each principal component $V_k$ is given by

$$V_k = \mathbf{v}^{(k)} \cdot \mathbf{q} = v_1(k) \cos\gamma_1 + v_2(k) \sin\gamma_1 + ... + v_{2N-1}(k) \cos\gamma_N + v_{2N}(k) \sin\gamma_N \qquad (2)$$

Where $\mathbf{v}^{(k)}$ is the $k$th eigenvector and $\{\gamma_n\}, n = 1, ..., N$, is the sequence of dihedral angles $(\phi, \psi)$ of the peptide backbone.

A measure of the influence of angle $\gamma_n$ on the principal component $V_k$ may be defined as

$$\Delta n(k) = (v_{2n-1}(k))^2 + (v_{2n}(k))^2 \qquad (3)$$

The length of each eigenvector is 1, and thus $\Sigma_n \Delta n(k) = 1$. $\Delta n(k)$ can hence be considered as the percentage of the effect of the angle $\gamma_n$ on the principal component $V_k$ [35]. These contributions per dihedral angle were calculated for the first principal component ($\Delta n(1)$) for both ATP1B1 and ATP1B2.

## Results

### Molecular modeling of human ATP1B1 and ATP1B2

The $\beta$-subunit of the sodium pump is a membrane protein with a single transmembrane helix and most of the mass folded as a Ig-like $\beta$-sandwich at the extracellular space [16,22]. Since the structure of the extracellular domain is stable and active [14,16,37–39], we decided to analyze its adhesive properties without the cytoplasmic and transmembrane domains as they do not participate in the $\beta$-$\beta$ trans-interaction. The identity between human $\beta_1$ subunit (P05026) and wild boar $\beta_1$ subunit (3WGU) is 92.41%. As we were interested in studying the intermolecular interactions, we had to consider that the 7.6% difference in sequence could result in a different behavior of protein-protein interactions. Therefore, it was important to work with a structural model. The three dimensional (3D) model of the extracellular domain of human Na$^+$, K$^+$-ATPase $\beta_1$ subunit (ATP1B1) was built by considering the crystal structure of the Na$^+$, K$^+$-ATPase 3WGU from wild boar (Sus scrofa) and the Fasta Sequence of Uniprot (P05026). When we compare the 3D structure of those two proteins, (the model for human $\beta_1$ and wild boar $\beta_1$) we find a homology of 95.5%. In Fig 1A the 3D model of the extracellular domain of $\beta_1$ subunit is depicted, considering residues 63 to 303. The three N-glycosylation sites: Asn158, Asn193 and Asn265 at the surface of the extracellular domain, the three disulfide bridges and the characteristic Ig-like $\beta$-sandwich structure are shown. Validation of the 3D model was carried out by employing Ramachandran plots, where it could be seen that 99% of the residues are included in permitted zones of the protein (Fig 1B). Since no crystal structure was available for the $\beta_2$ subunit of any species, the 3D model of the extracellular domain of human ATP1B2 was built by considering the crystal structure of the homologous (Identity: 40% and Convergence: 98%) pig gastric H+,K+-ATPase- 5YLU and the Fasta

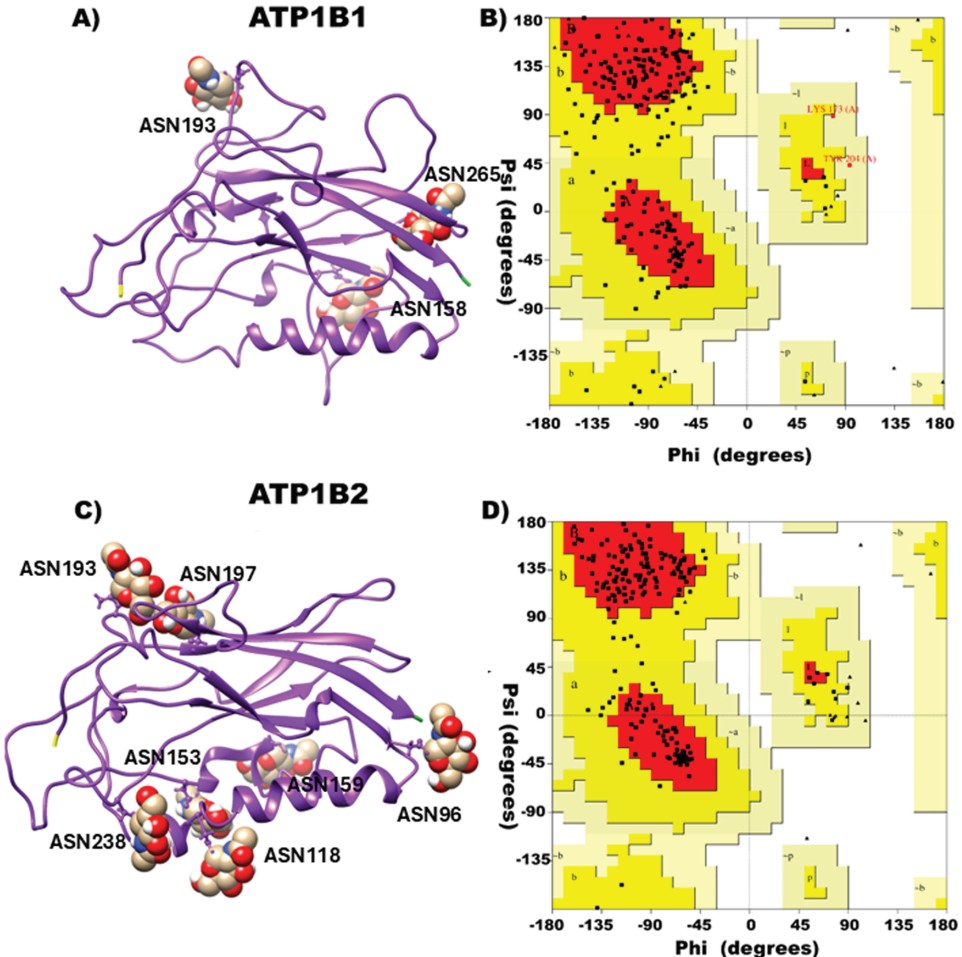

**Fig 1. Three-dimensional structure of monomeric $\beta_1$ subunit (ATP1B1) and $\beta_2$ subunit (ATP1B2).** A) 3D structure of the $\beta_1$ subunit monomer including its glycosylations. B) Ramachandran plot of the $\beta_1$ subunit monomer where it can be seen that only 1% of the residues of the proteins are included in the disallowed regions. C) 3D structure of the $\beta_2$ subunit monomer including its glycosylations. D) Ramachandran plot of the $\beta_2$ subunit monomer where it can be seen that none of the residues of the proteins are included in the disallowed regions. The extracellular domains of the $\beta_1$ and $\beta_2$ subunits depicted in A and C are with the N-terminal (residues 63 and 70) colored in yellow and the C-terminal (residues 303 and 289) in green. The distinctive alpha-helix of the $\beta$-subunits is localized in the bottom. For relative orientation to the alpha subunits, see S7 Fig.

Sequence of Uniprot (P14415). In Fig 1C, the following structural features of the extracellular domain (residues 70 to 289) of $\beta_2$ subunit are shown: seven N-glycosylation sites, three disulphide bridges and a characteristic Ig-like $\beta$-sandwich structure. Validation of the 3D models was carried out by employing Ramachandran plots, where it could be seen that 100% of the residues are included in permitted zones of the protein (Fig 1D).

## Building of the dimers $\beta_1 - \beta_1$ and $\beta_2 - \beta_2$

The molecular docking of the extracellular domains of both $\beta_1 - \beta_1$ and $\beta_2 - \beta_2$ subunits was performed by using HDOCK Server. In that protein-protein docking process, the most energetically favorable conformers were chosen, for $\beta_1 - \beta_1$ that of -193.04 kcal/mol and for $\beta_2 - \beta_2$

that of -274.99 kcal/mol) by means of the HDOCK Server. Among the favorable conformers, as a second structural criteria, we considered only the trans-dimers for further analyses. In the present work we considered pertinent to include the glycosylations in modeling, docking and MD simulations since it was demonstrated that N-glycosylation of both extracellular domains of $\beta_1$ and $\beta_2$ subunits are crucial for cell-cell adhesion [16,24]. In Fig 2 the selected trans-dimers were depicted.

## Molecular Dynamics Simulations of the dimers $\beta_1 - \beta_1$ and $\beta_2 - \beta_2$

Molecular dynamics simulations (MDS) were carried out on both dimeric complexes depicted in Fig 2, and trajectories were run for 200 ns. Furthermore, structural analysis was done with the Carma Program as described in "Methods". In agreement with previous results with dog ATP1B1 [16], the RMSD values for the soluble ectodomain of human $\beta_1 - \beta_1$ are within the range of 6-8Åeven though, the present model includes the three glycosylated residues. Fig 3 shows that there are no apparent structural differences between $\beta_1 - \beta_1$ and $\beta_2 - \beta_2$ dimers. Nevertheless, the surface residues that constitute the adhesion interface in the two dimers were different. Therefore, we decided to analyze both interfaces to get a better understanding about their formation and stability.

## Analysis of interactions in the $\beta - \beta$ interfaces

Protein-protein interactions are important for normal biological processes since they play a key role in the regulation of cellular functions that affect gene expression and function [40]. In this work we present an analysis of the residues at the interface of protein-protein interaction, thus providing information about the stability and specificity of the complex. In the analyses of the interfaces, the properties to be considered include: hydrogen bonding, buried surface area and hydrophobicity among others [41]. PPCHEK server was employed to get an insight on the non-bonded interactions that are present in the dimeric complexes ($\beta - \beta$)

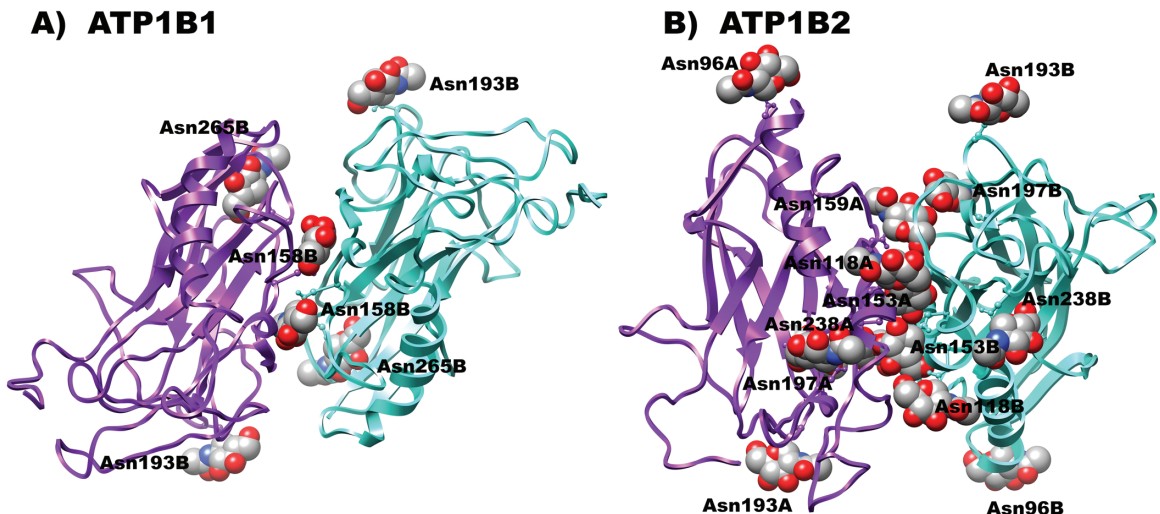

**Fig 2. Dimeric 3D structure of $\beta_1$ subunit and $\beta_2$ subunit in trans orientation.** A) Dimeric structure of $\beta_1 - \beta_1$. B) Dimeric structure of $\beta_2 - \beta_2$. For both cases Chain A is colored in purple and Chain B is colored in blue. Glycosylations are marked in balls and sticks.

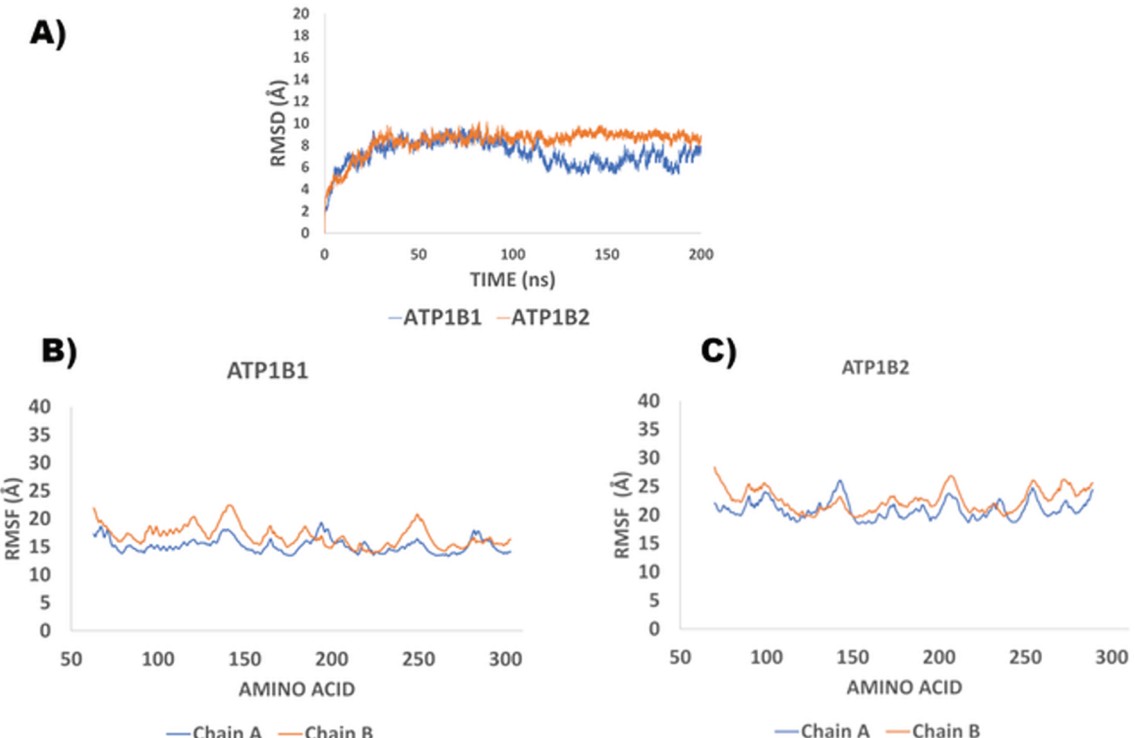

**Fig 3. Structural Analysis of ATP1B1 and ATP1B2 dimers: A) Root mean square deviation analysis (RMSD) of the dimers $\beta_1$ and $\beta_2$.** A) Root mean square deviation analysis (RMSD) of the dimers $\beta_1$ and $\beta_2$ . Root mean square fluctuation (RMSF) analysis of the alpha carbons of the dimers $\beta_1 - \beta_1$ (B) and $\beta_2 - \beta_2$ (C).

obtained from the MD simulations. These interactions in KJ/mol include: Hydrogen bonds, electrostatic energy, Van der Waals energy, and total stabilizing energy. PPCheck, can also predict reliably the correct docking pose by checking if the normalized energy per residue falls within a standard energy range of -2kJ/mol to -6kJ/mol which was obtained by studying a large number of well characterized protein-protein complexes [30]. Additionally, the percentage of residues in the interface of each of the dimers, at the different conformations, was investigated using the PDB-PISA server. Analysis of the following conformations: 0, 20, 60, 100, 120, 160 and 170 ns, was carried out employing the mentioned servers and are summarized in Tables S1 Table and S2 Table. A general observation is that the number of interface residues in the different conformations of $\beta_1$ dimers vary from that of $\beta_2$ dimers. This difference tends to be remarkable in the earlier protein conformations of the molecular dynamics simulations (S1 Table). The majority of the conformations (0, 20, 60, 100, 120 ns) of $\beta_1$ dimers show lower stabilizing energy in comparison to the conformations for $\beta_2$ dimers. The normalized energy per residue is around -2 kJ/mol in various conformers of $\beta_1 - \beta_1$ while in $\beta_2 - \beta_2$ none of the conformers reached that value. Thus, suggesting that in general, $\beta_1$ dimer is more stable in comparison to $\beta_2$ dimer (S2 Table). Even though selected snapshots were useful to depict structural differences in the distinct protein conformations, they seem not to reflect dynamics characteristics of both interfaces. Therefore, we used other tools in order to analyze and compare the dimeric interfaces.

## Searching for hot spots within dimeric interfaces

Protein-protein interactions in the interfaces were calculated through PDBsum software. Fig 4 depicts protein-protein interactions of the conformers of $\beta_1 - \beta_1$ and $\beta_2 - \beta_2$ taken at different times: 0,100,120 and 160 ns. The residues in the interface are depicted and some residues show to be constant in the interface of most of the conformers, for $\beta_1 - \beta_1$: Lys173, Gly225, Asn226, Glu228, Thr264, Leu266 and for $\beta_2 - \beta_2$: Arg130, Thr155, Ile163, Asn220 which are therefore considered the hot spots residues. The analysis of hot spot residues in each dimer suggests significant differences in the interfaces involved in homophilic protein-protein interactions between the $\beta_1$ and $\beta_2$ subunits of Na⁺, K⁺-ATPase across neighboring cells.

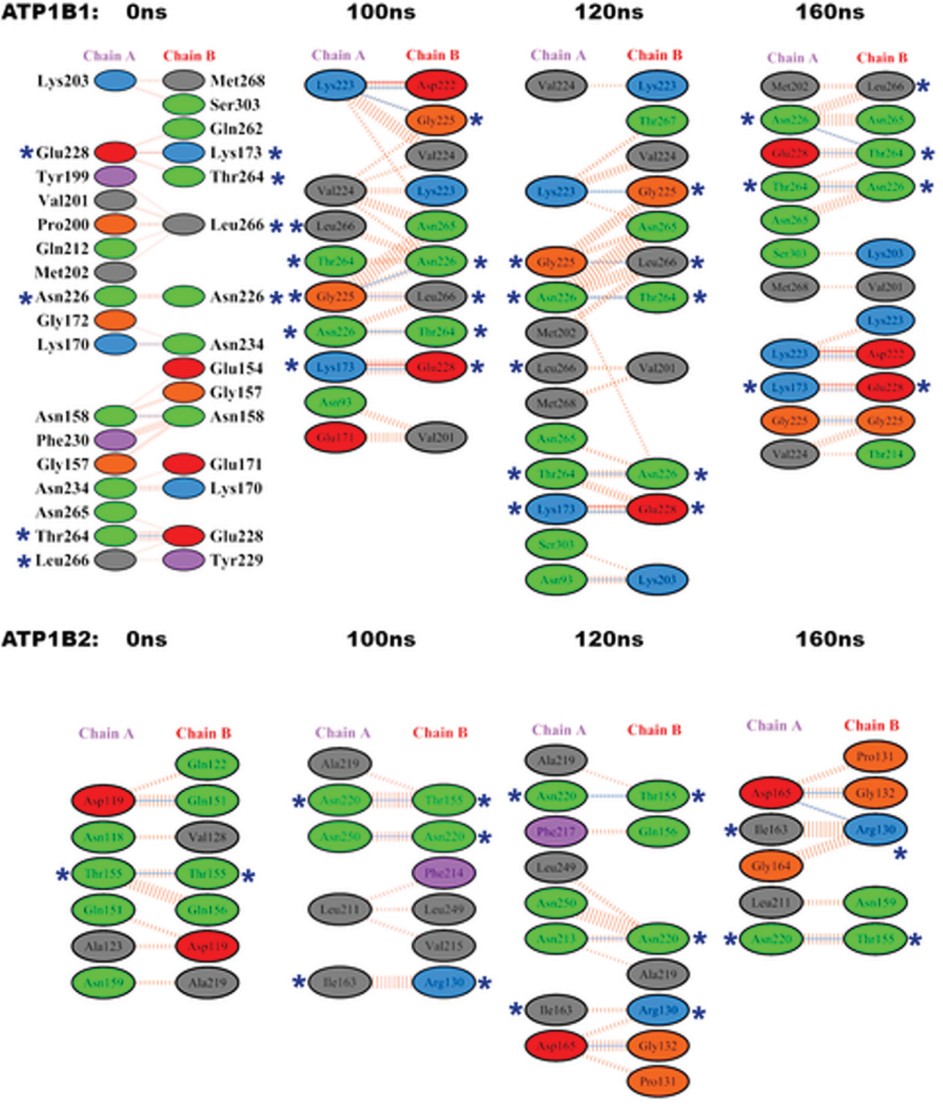

**Fig 4. Interfaces at ATP1B1 and ATP1B2 dimers.** Graphical representation of the protein interfaces at $\beta_1$ and $\beta_2$ dimers in different snapshots obtained from the MD simulations using PDBSUM server. (*) Residues that appear in all conformations are marked as hot spots.

The multiple sequence alignment presented in Fig 5 indicates that, despite high homology between the two subunits, the surface regions engaged in trans-dimerization differ. Notably, the hot spot residues of the $\beta_1$ dimer are clustered in close proximity, while those of the $\beta_2$ dimer are more widely dispersed across the surface. In this alignment, we compared the sequences of the dog ATP1B1 interface, as described in references 15 and 16, with those of human ATP1B1 and ATP1B2 examined in this study. Our findings reveal that: i) $\beta_2$ lacks segment 1 present in both dog and human $\beta_1$ (indicated by the orange box); ii) human $\beta_1$ shares hot spot residues with dog $\beta_1$ (highlighted in the green box); iii) residues Glu228, Lys173, Thr264, and Leu266 are conserved in both ATP1B1; and iv) residues Gly225 and Asn226 are identical across the three sequences. In relation to the hot spot residues of ATP1B2, we found that i) Arg130, Thr155, and Asn220 exhibit a lack of conservation; ii) although Ile163 is identical across the three sequences, it is exclusive to the $\beta_2 - \beta_2$ interface. Additionally, while human ATP1B2 shares Gly225 and Asn226 within segment 2, these residues do not seem to influence the $\beta_2 - \beta_2$ interface. Collectively, these structural distinctions provide insights into the observed differences between the dimer interfaces of $\beta_1$ and $\beta_2$.

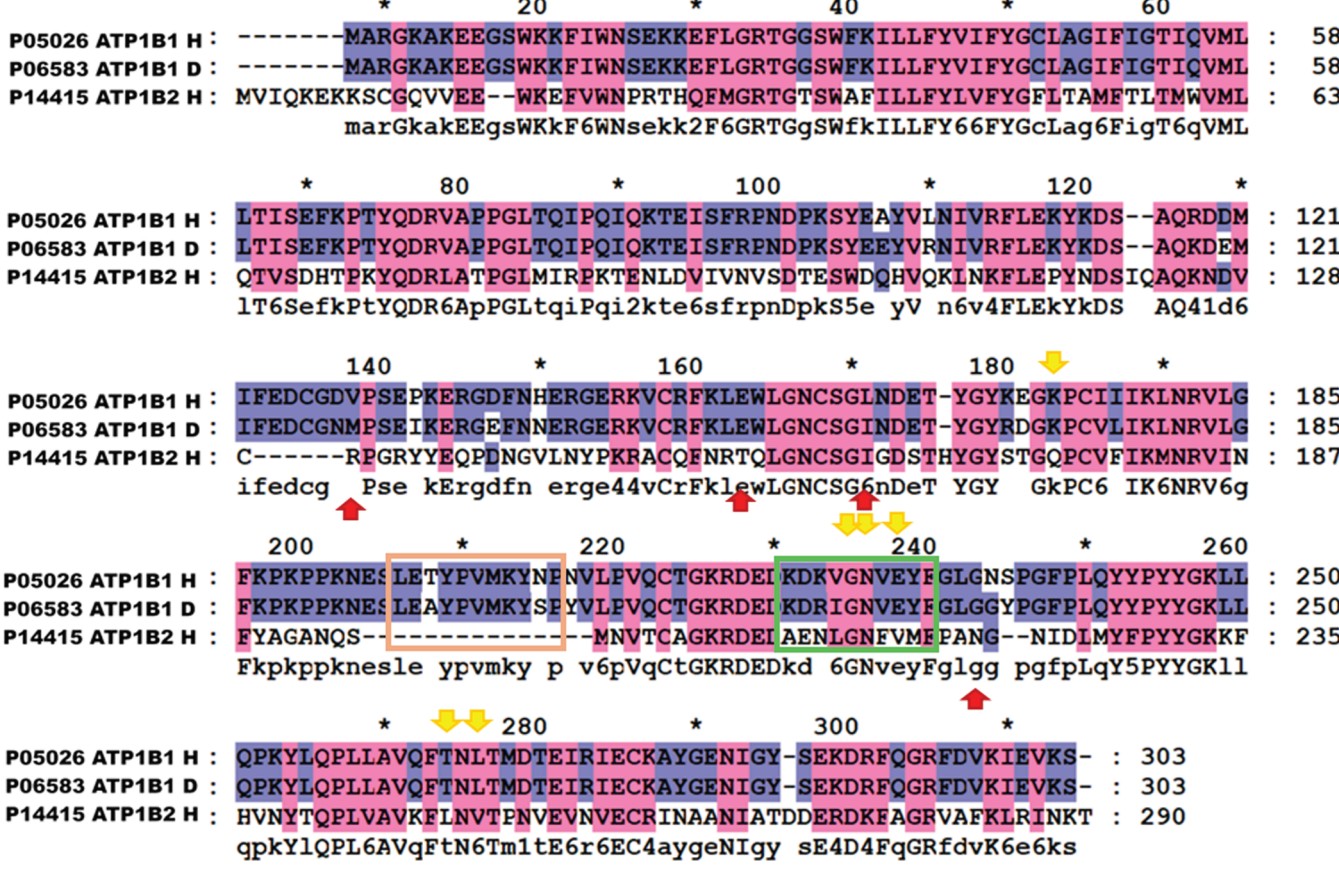

**Fig 5. Multiple sequence alignment of ATP1B1 and ATP1B2.** A multiple alignment is shown (ATP1B1 Human (P05026), ATP1B1 Dog (P06583), and ATP1B2 Human (P14415). Yellow Arrows: Hot Spots residues of human ATP1B1. Red arrows: Hot spots residues of human ATP1B2. Orange box: Dog Sequence 1 from ref. 15 and Green box: dog Sequence 2 from ref. 16.

## Monitoring interactions that involve the hot spot residues

Fersht and coworkers provided valuable information regarding the role of hydrogen bonds in protein stabilization [42]. Afterwards, several experimental studies were carried out on proteins of different nature, for example: BPTI [43], RNase Sa [44], Staphylococcal nuclease [45], human lysozyme [46]. In this work, one of the aims was to get insights about hydrogen bonds that are located in $\beta_1 - \beta_1$ and $\beta_2 - \beta_2$ dimers. For the case of $\beta_1$, from the three hot spot residues we could identify the formation of hydrogen bonds between the residues: Asn226A and Thr264B (Fig 6A). On the other hand, the hot spot residues in $\beta_2 - \beta_2$ identified as forming hydrogen bonds are Asn220A and Thr155B. These Hydrogen bonds were monitored along the trajectories of both $\beta_1 - \beta_1$ and $\beta_2 - \beta_2$ dimers. Fig 6A describes a constant hydrogen bond between residues Asn226A and Thr264B in $\beta_1$ dimer, since the first 20 ns of simulation. Fig 6B shows the formation of the hydrogen bond between Asn220A and Thr155B in $\beta_2$ dimer just after 100 ns of simulation. These results indicate a clear difference in the dynamics of the dimers formation and strongly suggest that these hydrogen bonds play a key role in the stabilization of the dimer.

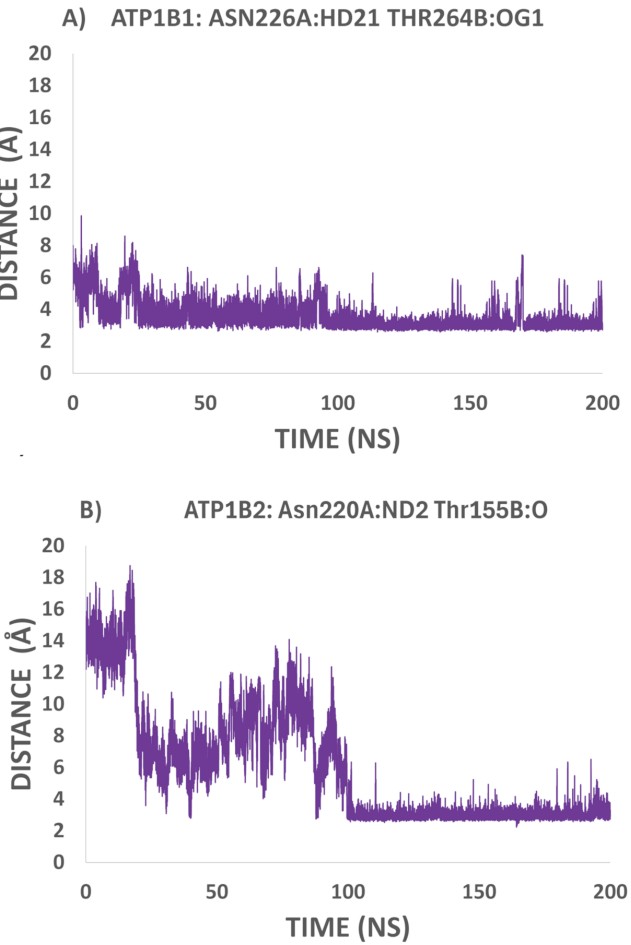

**Fig 6. Constant hydrogen-bonds in $\beta_1$ and $\beta_2$ dimers.** Distance between atoms (Asn226A:HD21-Thr264B:OG1) for $\beta_1$ and (Asn220A:ND2-Thr155B:O) for $\beta_2$ were calculated along the trajectories using Carma Software.

## Participation of N-glycosylated residues in the dimeric interface

N-Glycosylation involves adding oligosaccharides to the nitrogen atom of Asn in the Asn-X-Ser/Thr sequence of glycoproteins. This type of glycosylation is prevalent in many human proteins and is important for protein folding and stability of the protein [47], and targeting specific cellular locations [48,49]. All eukaryotic N-glycans share a common core of two N-acetylglucosamine (GlcNAc) and three mannose residues, which are further modified into diverse structures. Based on protein and cell type, N-glycans are classified as high-mannose, complex, or hybrid oligosaccharides. They regulate protein stability, solubility, trafficking, and cell signaling [47–50], Notably, human $\beta_1$ and $\beta_2$ isoforms differ in their N-glycosylation sites, with $\beta_1$ having three and $\beta_2$ seven. N-glycosylation is crucial for $\beta – \beta$ interactions, as its inhibition disrupts cell adhesion [24,49,51]. In our models for human $\beta_1$ and $\beta_2$ subunits, we introduced only the GlcNAc into the corresponding asparagines. Here, we identified the N-glycosylation sites (GlcNAc) located in the interfaces of $\beta_1 – \beta_1$ and $\beta_2 – \beta_2$ and their interactions with the surrounding residues at different conformers obtained from the MD simulations (0, 20, 60, 100, 120, 160, 170 ns). Tables Table 1 and Table 2 summarize the interacting glycans of N-glycosylated residues in each dimer and are centered in the table and labeled in red. For the case of the glycosylated Asparagines in $\beta_1$ dimer (Table 1), all the identified interactions are intramolecular, with residues of the same chain and are mainly through Van der Waals interactions. The most frequent intramolecular interactions with the glycosylation of Asn265 is through Van der Waals interactions, although few intramolecular hydrogen bonds were identified within the B chain (Thr267B y Thr270B). Noteworthy is the interaction with Thr264 considered a hot spot residue in $\beta_1 – \beta_1$ interface.

Table 2 shows the most frequent interactions with the glycans in the interface of the dimer $\beta_2 – \beta_2$. Two glycosylated residues, Asn118 and Asn197, show few interactions, mainly intramolecular ones. The other two are more interactive. Asn153A interacts with residues of the same chain mainly through Van der Waals interactions. Nevertheless, Asn153B interacts

**Table 1. Glycan-protein interactions of ATP1B1 ($\beta_1$ dimer).**

| N-linked glycosylation in ATP1B1: Asn158, Asn193 and Asn265 | | | | | | | | | | | | | | | | |
|---|---|---|---|---|---|---|---|---|---|---|---|---|---|---|---|---|
| Residue | Conformations (ns) | | | | | | | Residue | Conformations (ns) | | | | | | |
| | 0 | 20 | 60 | 100 | 120 | 160 | 170 | | 0 | 20 | 60 | 100 | 120 | 160 | 170 |
| GlcNAcAsn158A | | | | | | | | GlcNAcAsn158B | | | | | | | |
| Glu154A | | | ■ | ■ | ■ | ■ | ■ | Glu154B | | | ■ | ■ | ■ | ■ | ■ |
| Trp155A | | | ■ | ■ | ■ | ■ | ■ | Trp155B | | | ■ | ■ | ■ | ■ | ■ |
| Gly157A | | ■ | ■ | ■ | ■ | ■ | ■ | Gly157B | | ■ | ■ | ■ | ■ | ■ | ■ |
| Asn158A | ■ | ■ | ■ | ■ | ■ | ■ | ■ | Asn158B | ■ | ■ | ■ | ■ | ■ | ■ | ■ |
| Phe230A | | | ■ | ■ | ■ | ■ | | | | | | | | | |
| GlcNAcAsn193A | | | | | | | | GlcNAcAsn193B | | | | | | | |
| Asn193A | ■ | ■ | ■ | ■ | ■ | ■ | ■ | Asn193B | ■ | ■ | ■ | ■ | ■ | ■ | ■ |
| GlcNAcAsn265A | | | | | | | | GlcNAcAsn265B | | | | | | | |
| Phe263A | ▣ | ■ | ■ | ■ | ■ | ■ | ■ | Val224B | ■ | ▣ | ■ | ■ | ■ | ■ | ■ |
| Thr264A | | ■ | ■ | ■ | ■ | ■ | ▨ | Phe263B | ▣ | ■ | ■ | ■ | ■ | ■ | ▣ |
| Asn265A | ▣ | ■ | ■ | ■ | ■ | ■ | ■ | Thr264B | | ■ | ■ | ■ | ■ | ■ | ■ |
| Ile272A | ▣ | ■ | ■ | ■ | ■ | ■ | ■ | Asn265B | ■ | ■ | ■ | ■ | ■ | ■ | ■ |
| | | | | | | | | Thr267B | ■ | ▣ | ▣ | ▣ | ▣ | ▣ | ▣ |
| | | | | | | | | Thr270B | ▣ | ■ | ▣ | ▣ | ▣ | ▣ | ▣ |
| | | | | | | | | Ile272B | ■ | ■ | ■ | ▣ | ▣ | ■ | ■ |

Different conformations of the protein are shown as rectangles (0, 20, 60, 100, 120, 160 and 170 ns). Interactions of the glycans are marked with different colors as follows: Van der Waals (green), Hydrogen bonds (blue) and Carbon-Hydrogen (purple).

**Table 2. Glycan-protein interactions of ATP1B2 ($\beta_2$ dimer).**

| N-linked glycosylation in ATP1B2: Asn118, Asn153, Asn159 and Asn197 | | | | | | | | | | | | | | | |
|---|---|---|---|---|---|---|---|---|---|---|---|---|---|---|---|
| Residue | Conformations (ns) | | | | | | | Residue | Conformations (ns) | | | | | | |
| | 0 | 20 | 60 | 100 | 120 | 160 | 170 | | 0 | 20 | 60 | 100 | 120 | 160 | 170 |
| GlcNAcAsn118A | | | | | | | | GlcNAcAsn118B | | | | | | | |
| Asn118A | | | | | | | | Asn118B | | | | | | | |
| GlcNAcAsn153A | | | | | | | | GlcNAcAsn153B | | | | | | | |
| Asp119A | | | | | | | | Asp119B | | | | | | | |
| Gln122A | | | | | | | | Asn153B | | | | | | | |
| Ala123A | | | | | | | | Thr155A | | | | | | | |
| Gln151A | | | | | | | | Gln156A | | | | | | | |
| Thr155A | | | | | | | | Phe217A | | | | | | | |
| GlcNAcAsn159A | | | | | | | | GlcNAcAsn159B | | | | | | | |
| Val128B | | | | | | | | Asn118A | | | | | | | |
| Arg130B | | | | | | | | Arg154A | | | | | | | |
| Asn159A | | | | | | | | Asn159B | | | | | | | |
| Asp165A | | | | | | | | | | | | | | | |
| Met226B | | | | | | | | | | | | | | | |
| GlcNAcAsn153A | | | | | | | | | | | | | | | |
| GlcNAcAsn197A | | | | | | | | GlcNAcAsn197B | | | | | | | |
| | | | | | | | | Asn193B | | | | | | | |

Different conformations of the protein are shown as rectangles (0, 20, 60, 100, 120, 160 and 170 ns). Interactions of the glycosylation are marked with different colors as follows: Van der Waals (green), Hydrogen bonds (blue) and Carbon-Hydrogen (purple).

with residues of the contrary chain through Van der Waals and hydrogen bonds; a similar behavior is observed with Asn159A and Asn159B. Of worthy interest, residues Arg130B and Thr155A which we identified as hot spots within $\beta_2 - \beta_2$ interface, interact with the glycans of Asn159A and Asn153B, respectively.

## Prediction of Binding Free energy through MM-PBSA method

Here we present an easy-to-use pipeline tool named Calculation of Free Energy (CaFE) to conduct Molecular Mechanics Poisson-Boltzmann Surface Area (MM-PBSA) and LIE calculations. Powered by the VMD and NAMD programs, CaFE is able to handle numerous static coordinate and molecular dynamics trajectory file formats generated by different molecular simulations. The MM-PBSA approach has been widely applied as an efficient and reliable free energy simulation method to model molecular recognition, such as for protein-ligand binding interactions [33]. Moreover, MM-PBSA and MM-GBSA methods are useful methods in both accuracy and computational effort between empirical scoring and strict alchemical perturbation methods [52]. Binding free energy of the dimer complexes $\beta_1$ and $\beta_2$ was calculated by CaFE to conduct MM-PBSA [32] and the obtained values are presented in Table 3. It can be seen that the major contribution to the free energy of the complex is due to polar interactions. Dimeric complex of $\beta_2$ shows higher binding free energy in comparison to the dimeric complex of $\beta_1$ (–19707.5 vs –22671.13 kcal/mol) .

## Principal component analysis (PCA)

In Silico approaches are useful to describe protein dynamics, in which fluctuations range from bond-distance variations. Molecular dynamics simulations along with mathematical applications are very helpful to investigate these fluctuations that occur in the proteins. Principal component analysis (PCA) is a useful mathematical technique to reduce a multidimensional

**Table 3. MM-PBSA calculation of ATP1B1 ($\beta_1$) and ATP1B2 ($\beta_2$) dimeric complexes.**

| Protein | Electro-static (kcal/mol) | Van der Waals (kcal/mol) | Poisson-Boltzmann (kcal/mol) | Surface Area (kcal/mol) | Gas (kcal/mol) | Solvate (kcal/mol) | Polar inter-actions (kcal/mol) | Non Polar interactions (kcal/mol) | Total (kcal/mol) |
|---|---|---|---|---|---|---|---|---|---|
| ATP1B1 | −11375.42 | −1664.37 | −9782.35 | 151.007 | −13039.79 | −9631.34 | −21157.77 | −1513.361 | −22671.1333 |
| ATP1B2 | −9546.50 | −1514.63 | −8784.48 | 138.04 | −11061.14 | −8646.44 | −18330.98 | −1376.600 | −19707.5810 |

complex set of variables to a lower dimension. This technique has been used to investigate the stages of protein folding in proteins of diverse nature [53]. In general, the great majority of proteins show particular behavior in which their two/three principal components describe the main motions of the proteins (about 70-80%). As we can infer from our results, the cumulative contribution to the variance in the conformational space is the largest for the first two principal components, 50% and 30% for $\beta_1$ and $\beta_2$ respectively (S1 Fig). Dihedral angle principal component analysis (dPCA) has shown advantages for the treatment of proteins and was therefore used for this study. We studied the fluctuations of these principal components (PC1 and PC2). Projection of the trajectories onto PC1 and PC2, together with the cluster analysis (where the region with the highest density is highlighted as cluster 1) is depicted in S2 Fig. The free energy landscapes obtained from the dPCA analysis (Fig 7) show the region with the highest density as the deepest basin ($\Delta G = 0$). We observed that $\beta_1$ homodimer presents low values for PC1 (around 0) and high values for PC2 (around 5) in its region with the highest density, which is rather localized. On the other hand, the region with the highest density for $\beta_2$ homodimer spans a large region where values for PC1 range from -6 to -2 and values for PC2 range from -4 to 1. Having a high density region with the largest principal component close to zero as is the case for $\beta_1$ dimer suggests the formation of a stable interface that has a short-ranged oscillation. On the other hand, large absolute values in the largest principal component of the region with the highest density, as observed for $\beta_2$ dimer, suggest that under these conditions a stable dimer is not yet reached. The motions associated with PC1

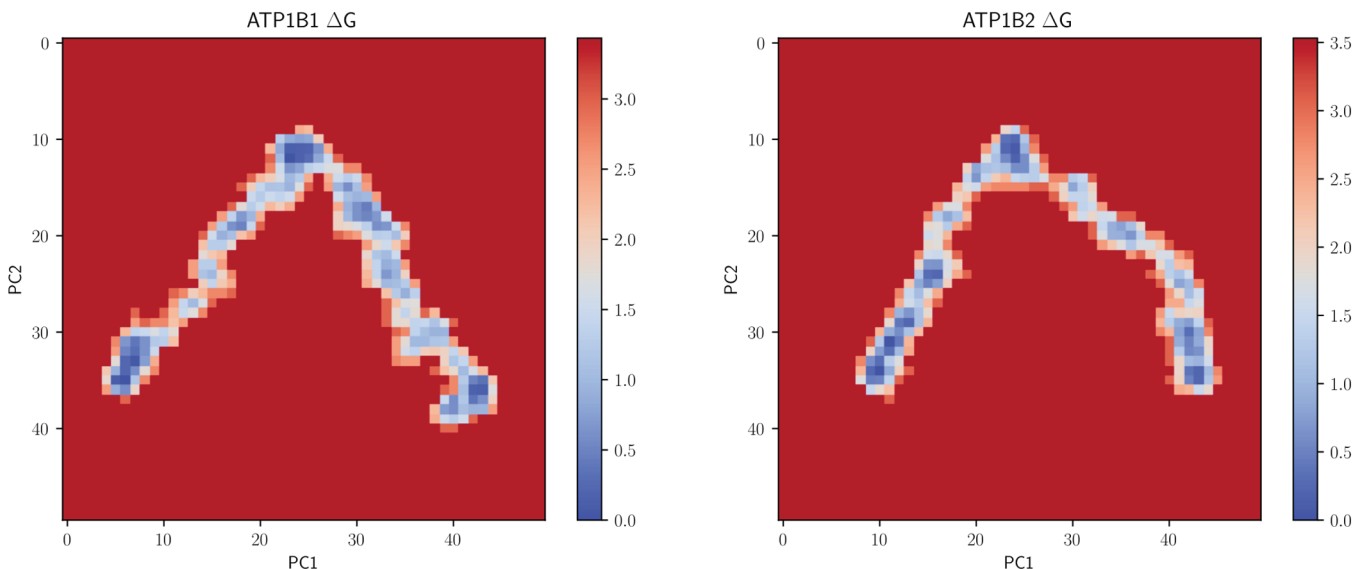

**Fig 7. Free energy landscapes considering the first two principal components of $\beta_1 - \beta_1$ and $\beta_2 - \beta_2$ dimers.**

and PC2 for $\beta_1$ dimer both show symmetric, rotatory behaviors, whereas, PC1 for $\beta_2$ dimer shows a longitudinal motion and PC2 seems to involve substantially the most mobile loops; this participation of the loops in PC2 suggests that a concerted motion involving the interface has not been reached for this complex (Fig 8, S3 Fig to S7 Fig and Supplementary Videos).

### Analysis of the movement contributions per dihedral angle

In the dPCA, each dihedral angle $\gamma$ is transformed into a space with two coordinates ($\cos\gamma$, $\sin\gamma$). Each principal component has a weight calculated for each of those coordinates and a measure of the influence of angle $\gamma$ on principal component $k$ ($\Delta\gamma(k)$) is defined as the sum of the squares of the corresponding weights, as detailed in section 4.8. The contribution to the first principal component from every angle ($\Delta(1)$) was calculated (S8 Fig). For $\beta_1$, elevated contributions are observed in the vicinity of interface residues Val129, Pro130, Glu132 and Pro133 in Chain A, and in the vicinity of interface residues Glu165, Thr166, Asp218 and Asp220 in Chain B. Thus, all interface residues are close to regions with a large contribution and are therefore participating in the main motion of the complex. These observations suggest a stable interface for this complex. On the other hand, the motion in $\beta_2$ is quite asymmetrical, while there are major peaks for interface residues Asn193 and Ala265 in both Chain A and Chain B, interface residues Met216, Ala219, Asn220, Gly221, Asn222, Ile223, Asp224 and Lys234 are in a region with a rather small contribution in Chain A and regions with small to negligible contributions in Chain B. Interface residue Gly158 is in a region with almost zero contribution in both Chain A and Chain B. The contribution of residue Asn197A in $\beta_2$ to the motion is high, whereas Asn197B does not move significantly, probably due to other interactions that restrain this movement. These findings suggest that the interface is not part of the main motion of this complex and is thus not likely to have reached a stable state.

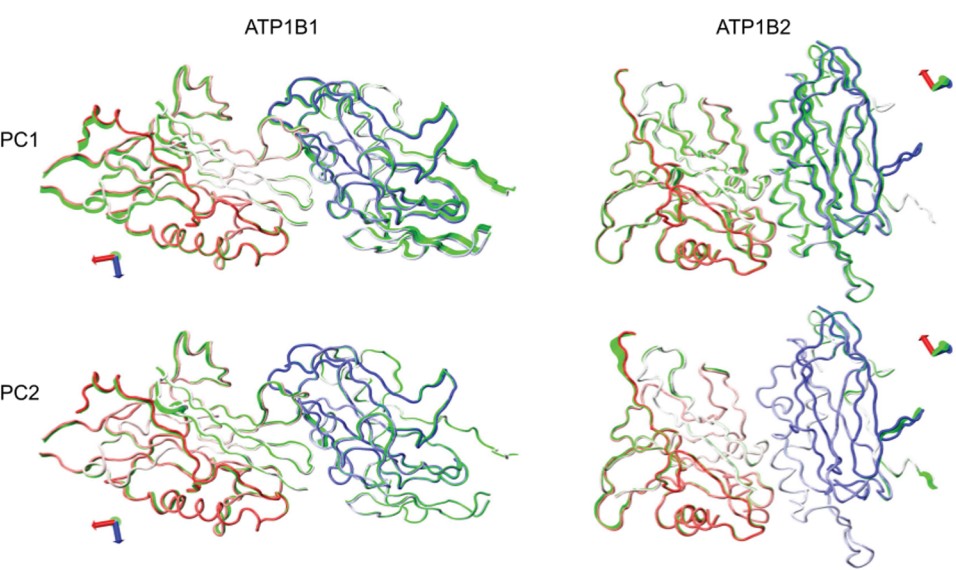

**Fig 8. Motion associated with PC1 and PC2.** Chain A goes from red in the N-terminus to white in the C-terminus while chain B goes from white in the N-terminus to blue in the C-terminus. Green tubes show the motion associated with the principal components.

## Discussion

Our in silico investigations yield significant insights into the structural dynamics underlying the trans-dimerization process of the extracellular domains of human Na$^+$, K$^+$-ATPase $\beta$-subunits, namely ATP1B1 ($\beta_1$) and ATP1B2 ($\beta_2$). Previous works have individually studied the structural features of Dog $\beta_1$ [15,16,37] and human $\beta_2$ [24], revealing notable molecular and biological distinctions in their adhesive properties. In the current study, we expand upon the analysis of $\beta_1$ and $\beta_2$, employing docking and molecular dynamics (MD) simulations and leveraging in silico methodologies to investigate various structural aspects. Our aim is to elucidate the biological disparities observed between $\beta_1$ and $\beta_2$/AMOG subunits as adhesion molecules.

### Analysis of the interacting interfaces of $\beta_1 – \beta_1$ and $\beta_2 – \beta_2$ dimers along the MD trajectories

Initially, we examined the interacting interfaces of both $\beta_1$ – $\beta_1$ and $\beta_2$–$\beta_2$ dimers, revealing a consistent reduction in the number of residues within the interface of $\beta_2$ dimer throughout the simulation compared to $\beta_1$ dimer (see Fig. 4). This reduction in residue count corresponds to a lower interface area in $\beta_2$ dimer, which correlates with reduced complex stability attributable to weaker interactions within this region (Table 5). Furthermore, we identified hot spot residues pivotal for interface stabilization. Notably, the number of hot spot residues within the $\beta_2$ – $\beta_2$ interface was found to be lower than that within the $\beta_1$ – $\beta_1$ interface. The interface of the dimer of human $\beta_1$ – $\beta_1$ that represents the starting point for the molecular dynamic simulation (0 ns in Fig 4) is very similar to that of the dog $\beta_1$ – $\beta_1$ proposed by the in silico and in vitro analyses of [16]. Nevertheless, during the MD simulation the interactions at the interface become less ample and at least 5 residues are identified as hot spots localized in a domain that range between Gly225, and Leu266. On the other hand, the hot spots residues of $\beta_2$ dimer are more dispersed and indicate a large interface surface. The alignment of human and dog ATP1B1 and human ATP1B2 in Fig 5 shows that indeed the apparent hot spot residues are localized in very distant domains on $\beta_2$ dimer that do not overlap with those of $\beta_1$ – $\beta_1$ interface. This observation correlates with the MM-PBSA calculations (Table 5) , that shows lower binding free energy for $\beta_2$ dimer in comparison to $\beta_1$ – $\beta_1$ (-19707.5 vs -22671.13kcal/mol, respectively). Meticulous analyses of the first two principal components from the dPCA performed revealed significant differences in the dynamics of the $\beta_1$ and $\beta_2$ dimers. First, the most populated energy region for $\beta_1$ is rather localized whereas for $\beta_2$ the most populated energy region spans a large area. Besides, $\beta_1$ shows a symmetric, concerted rotation involving the interface residues in both monomers, whereas $\beta_2$ shows a tendency to increase the distance between the monomers in its main motion (PC1), while the second main motion does not involve significantly the interface residues. Additionally, for $\beta_1$, an important contribution is observed for the dihedrals surrounding Asn158 and Asn193. The former is close to interface residues Glu165 and Thr166 and its pronounced motion could influence this region of the interface. Interface residues Asp218 and Asp220 show important contributions, which suggests their involvement in the concerted motion of the interface. In contrast, most glycosylated asparagines and interface residues show minute contributions for $\beta_2$ dimer. All these observations are consistent with the existence of a stable interface in $\beta_1$ dimer and the lack thereof in $\beta_2$ dimer. Lee et al study, the impacts of N-glycans on the folded glycoproteins in terms of protein structure and dynamics in their glycosylated and deglycosylated forms using an integrated computational approach of the Protein Data Bank (PDB) structure analysis and atomistic molecular dynamics (MD) simulations [54]. This

study reveals that N-glycosylation does not induce significant global/local changes in protein structure, but decreases protein dynamics, likely leading to an increase in protein stability. Interestingly, for $\beta_2$ dimer Asn159A, Asn153B and Asn159B form favorable interactions with residues in the opposite chain (Table 4), which suggests that these glycosylations could play an instrumental role in keeping the dimer despite the motions that tend to separate the monomers.

## Biological divergence observed between $\beta_1$ and $\beta_2$/AMOG subunits

Through structural analysis, stability assessments, movement analyses, and free energy calculations, this study reveals that $\beta_2$, a conventional component of astrocytic Na$^+$-pumps, does not engage in $\beta_2 - \beta_2$ trans-dimerization among astrocytes [17], as also demonstrated by protein-protein interaction assays such as pull-down experiments [24]. In contrast, $\beta_2$ exhibits a propensity for trans-dimerization when expressed in CHO, MDCK, or U87-MG glioma cells [24,55]. These findings suggest the involvement of modulatory elements that promote stable $\beta_2 - \beta_2$ interactions in transfected cell lines but inhibit them in astrocytes, potentially masking the trans interaction capacity of $\beta_2$ in the latter context. Notably, the Na$^+$, K$^+$-ATPase complex in astrocytes, comprising $\alpha_2$ and $\beta_2$ subunits, has been identified as part of a functional assembly on the astrocytic plasma membrane [56]. This complex regulates lactate transport via coordinated interactions among GluR2, PrP, $\alpha_2$ , $\beta_2$, basigin, and MCT1. Within this assembly, $\beta_2$'s N-glycans (oligomannose) interact with the lectin domain of basigin [57]. As specific N-glycosylation sites of $\beta_2$ are suggested to be at the dimer interface, basigin-$\beta_2$ interaction would probably impair $\beta_2 - \beta_2$ trans-interactions. A relevant participant of cis-interactions with $\alpha_2$ $\beta_2$ would be the regulatory FXYD protein. Nonetheless, any FXYD protein was detected in that study. As reported recently [7] astrocytes express the FXYD1 (phospholemman) member of that family. Interestingly, FXYD1 stabilizes and protects from thermal inactivation the Na,K-ATPase in a mammalian cell membrane [58]. Thus, it is plausible that the interaction of FXYD1 with $\beta_2$ is an additional structural constraint that limits the $\beta_2 - \beta_2$ trans interactions between astrocytes. While we exclude from this analysis any potential interactions between the extracellular domains of $\beta_2$, the $\alpha_2$ -subunit, and FXYD1, the possibility of cis interactions involving surface residues within proteins at the same membrane remains to be explored. Future studies should address these gaps to provide a more comprehensive understanding of the structural and functional dynamics of $\beta_2$ in astrocytes. Of worthy interest is that glycan-protein interactions can be considered as multivalent interactions which are often required to achieve biologically relevant binding even though they are known to have low affinity [59]. On the other hand, these interactions have been related to some other functions which include: dynamic forms of adhesion mechanisms, for example, rolling (cells), stick and roll (bacteria) or surfacing (viruses) [59]. Glycosylations play a pivotal role in cell adhesion and recognition, and can also influence protein-protein interactions. Interestingly, the main difference between the $\beta_1$ and the $\beta_2$ isoforms is in their number and sites of N-glycosylation. While human $\beta_1$(ATP1B1) carries three conserved N-Glycosylation sites, $\beta_2$ (ATP1B2) conserves these three sites but has 4 additional ones. In the case of cell adhesion mediated by trans-interactions of $\beta_1 - \beta_1$ and $\beta_2 - \beta_2$, the N-glycosylation of both $\beta$-subunits had been reported to play an important role [15,24,49,51]. Here, in a detailed analysis of the interactions of the core-glycosylated residues, we observed that this type of interactions in $\beta_1$ dimer occur within residues located at the same chain; whereas $\beta_2$ dimer shows interactions that occur both intra- and inter-molecular, between contrary chains. Understanding the cellular physiology of $\beta_2$/AMOG is gaining renewed interest due to the increasing evidence implicating Na$^+$, K$^+$-ATPase in neurological pathologies and

disorders. Aberrant expressions of different Na$^+$, K$^+$-ATPase subunits and their activity have been linked to the development and progression of various cancers, as well as cancer cell proliferation, migration, and apoptosis [59]. However, the exact mechanism by which Na$^+$, K$^+$-ATPase influences cellular migration and invasion in cancer remains unclear. In the brain, several mutations and aberrant expressions of Na$^+$, K$^+$-ATPase $\alpha$ and $\beta$ isoforms have been associated with both neurological phenotypes [60] and brain cancer [61]. Remarkably, the majority of Glioblastoma multiforme (GBM) tumors exhibit a dramatic loss of $\beta_2$/AMOG expression. Sun et al. [61] proposed that this loss may be a key mechanism contributing to the increased invasiveness of GBM cells. They found that overexpression of $\beta_2$/AMOG reduced the invasion of GBM cells and brain tumor-initiating cells (BTICs) without affecting their migration or proliferation. Conversely, knockdown of $\beta_2$/AMOG expression in normal human astrocytes increased their invasiveness. Collectively, these findings implicate $\beta_2$/AMOG in glioma invasion, suggesting that downregulation of $\beta_2$/AMOG expression is a crucial step in the differentiation of BTICs. Therefore, $\beta_2$/AMOG is considered a tumor-suppressing protein and is of great interest for understanding its function in the central nervous system. Although our findings suggest that $\beta_2$ subunit can form homotypic trans-dimers, it does not exclude the proposal of the Schachner group of forming heterotypic interactions that regulate neurite outgrowth and cell migration during development [17–19]. Therefore, $\beta_2$ subunit on astrocyte plasma membrane is probably able to form a stable interface with a yet unknown neural receptor. Various published works appoint the participation of AMOG/$\beta_2$ subunit in signaling pathways [54,62]. In none of those works, the ligand that activates that signaling pathway was identified. Interestingly, Litan et al. report the participation of $\beta_2$ subunit in signaling pathways that involve Merlin and EGFR in neuronal granular cells [63]. Their model suggests $\beta_2 - \beta_2$ interaction as a switch to activate that pathway. Nevertheless, they do not discuss that point further. Our future work is directed to identify that heterotypic partner of $\beta_2$ on neurons and study their interaction.

## Conclusions

*In this study, we identify key structural features underlying the differences on homotypic adhesive functions between $\beta_1 - \beta_1$ and $\beta_2 - \beta_2$ complexes.* First, the interface composition is influenced by sequence and *structural variations* between the two isoforms. Second, surface glycosylation differs significantly, with $\beta_2$ exhibiting more N-glycans. While these glycans do not mediate protein-protein interactions in $\beta_1$, they appear essential for facilitating such interactions in $\beta_2$. However, the trans-dimer formed by $\beta_2$ subunits is not a stable complex, suggesting that a stable $\beta_2 - \beta_2$ interface may require additional cellular components or co-factors not accounted for in our current model.

## Supporting information

t**S1 Table. Interactions in the interface calculated in the different conformations of ATP1B1 and ATP1B2.**
(DOCX)

**S2 Table. Interactions in the interface calculated in the different conformations of ATP1B1 and ATP1B2 dimers.** The interaction is expressed as pseudo energy, whose ranges have been standardized using known sets of protein-protein complexes.
(DOCX)

**S1 Fig. Cumulative contribution of the principal components to the variance in the structural space.** The first principal components for both dimers explain under 40% of the energy observed in the simulation. Reaching 80% requires over 100 principal components, which

suggests the simulation time might have to be extended to allow fewer motions to dominate the dynamics. The projections of the trajectories onto PC1 and PC2 are very different, which was expected as the sequences show only partial similarity. Regarding the cluster analysis, some clusters of similar size to the main cluster are observed, which correlates with the poor dominance shown by the main principal components and hints at the possibility of a main energetic basin still waiting to be populated. The motions associated with the two main principal components for ATP1B1 show symmetric, rotatory behaviors that are expected in a stable dimer. In contrast, for ATP1B2, PC1 shows an asymmetric, longitudinal motion that seems to drive the monomers away from each other while PC2 seems to involve mostly inconsequential motions in the most mobile loops. This behavior can be related to Table I, where ATP1B2 shows unfavorable interactions for several of the conformations considered. (TIF)

**S2 Fig. dPCA Cluster analysis considering the first two principal components of $\beta_1$ – $\beta_1$ and $\beta_2$ – $\beta_2$ dimers.** For ATP1B1 we have the main cluster around low values of PC1, whereas the main cluster for ATP1B2 is in a region with large values for both PC1 and PC2, and it also spans a larger region. These observations support the conclusion that ATP1B1 shows a stable interface and the lack thereof for ATP1B2. (TIF)

**S3 Fig. Motion associated with ATP1B1 dimer, PC1, stereo image.** Chain A goes from red in the N-terminus to white in the C-terminus while chain B goes from white in the N-terminus to blue in the C-terminus. Green tubes show the motion associated with the principal components. The motion for PC1 is concerted, symmetric and rotatory, suggesting that a stable interface is reached in the simulation for this dimer. (TIF)

**S4 Fig. Motion associated with ATP1B1 dimer, PC2, stereo image.** Chain A goes from red in the N-terminus to white in the C-terminus while chain B goes from white in the N-terminus to blue in the C-terminus. Green tubes show the motion associated with the principal components. The motion for PC2 shows less amplitude than the motion for PC1, but is also concerted, symmetric and rotatory, suggesting that a stable interface is reached in the simulation for this dimer. (TIF)

**S5 Fig. Motion associated with ATP1B2 dimer, PC1, stereo image.** Chain A goes from red in the N-terminus to white in the C-terminus while chain B goes from white in the N-terminus to blue in the C-terminus. Green tubes show the motion associated with the principal components. The motion for PC1 is longitudinal instead of rotatory and shows a tendency to increase the distance between the monomers, which suggests that a stable dimer is not reached in the simulation. (TIF)

**S6 Fig. Motion associated with ATP1B2 dimer, PC2, stereo image.** Chain A goes from red in the N-terminus to white in the C-terminus while chain B goes from white in the N-terminus to blue in the C-terminus. Green tubes show the motion associated with the principal components. The motion for PC2 does not involve significantly the interface residues, which is consistent with the lack of a stable interface for this dimer. (TIF)

**S7 Fig. Motion associated with PC1 and PC2 of $\beta_1$ and $\beta_2$ dimers, with structural references.** Coloring of the dimers and the associated motions as in Fig. 8. The position of alpha subunit L7-8 (shown in yellow) and the position of $\beta$ subunit transmembrane span (shown in

cyan) for $\beta_1 - \beta_1$, and the predicted positions of L7-8 (shown in yellow) and the transmembrane span for $\beta_2 - \beta_2$ (shown in cyan) were structurally aligned from the crystal structures used for the modeling (3WGU and 5YLU, respectively). The groove that would accommodate L7-8 is preserved in all cases.
(TIF)

**S8 Fig. Analysis of the movement contributions per dihedral angle.** Vertical numbers indicate interface residues while horizontal numbers indicate glycosylated asparagines. For ATP1B1, Chain A, the highest peaks are in 5 $\Phi$ angles (Asn193, Asp222, Tyr254, Asp269, Arg290) and 9 $\Psi$ angles (Val72, Glu135, Arg136, Asp138, Phe139, Asn193, Lys253, Asp289, Arg290), while for Chain B the highest values are found in 10 $\Phi$ angles (Tyr68, Arg143, Gly144, Glu145, Ser160, Ser195, Lys216, Arg217, Asp222, Tyr254) and 15 $\Psi$ angles (Tyr68, Asp70, Arg71, Gln84, Asn93, Arg143, Glu145, Gly161, Gly172, Asn193, Lys221, Lys253, Lys288, Phe291, Gly293). The distribution of peaks for ATP1B2 is somewhat different, as for Chain A the highest peaks are in 8 $\Phi$ angles (Asn90, Leu91, Cys129, Arg133, Gln137, Asn193, Ala266, Asn267) and 17 $\Psi$ angles (Glu89, Asn90, Val128, Gly132, Glu136, Ala192, Met196, Asp205, Glu206, Tyr231, Asn264-Thr270) while for Chain B the highest values are in 3 $\Phi$ angles (Leu143, Phe188, Asp272) and 5 $\Psi$ angles (Gln74, Gly141, Leu143, Asn187, Tyr189).
(TIF)

**S1 Video. Motion associated with ATP1B1 dimer, PC1.**
(MP4)

**S2 Video. Motion associated with ATP1B1 dimer, PC2.**
(MP4)

**S3 Video. Motion associated with ATP1B2 dimer, PC1.**
(MP4)

**S4 Video. Motion associated with ATP1B2 dimer, PC2.**
(MP4)

## Acknowledgments

The authors would like to thank the staff of the Computing Cluster Xiuhcoatl-Cinvestav (granted by LANCAD) and Hector Manuel Oliver Hernandez.

## Author contributions

**Conceptualization:** Liora Shoshani, Marlet Martínez-Archundia.

**Data curation:** Gema Ramírez-Salinas, Marlet Martínez-Archundia.

**Formal analysis:** Gema Ramírez-Salinas, Jorge L. Rosas-Trigueros, Christian Sosa Huerta, Marlet Martínez-Archundia.

**Funding acquisition:** Gema Ramírez-Salinas, Marlet Martínez-Archundia.

**Investigation:** Liora Shoshani, Christian Sosa Huerta, Marlet Martínez-Archundia.

**Methodology:** Jorge L. Rosas-Trigueros, Marlet Martínez-Archundia.

**Project administration:** Liora Shoshani, Marlet Martínez-Archundia.

**Software:** Gema Ramírez-Salinas, Marlet Martínez-Archundia.

**Supervision:** Gema Ramírez-Salinas, Liora Shoshani, Marlet Martínez-Archundia.

**Validation:** Gema Ramírez-Salinas, Jorge L. Rosas-Trigueros, Marlet Martínez-Archundia.

**Visualization:** Liora Shoshani, Jorge L. Rosas-Trigueros, Marlet Martínez-Archundia.

**Writing – original draft:** Gema Ramírez-Salinas, Liora Shoshani, Jorge L. Rosas-Trigueros, Christian Sosa Huerta, Marlet Martínez-Archundia.

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
