## [Decision Letter · Decision Letter 0]

24 Oct 2024

PONE-D-24-40078In silico studies provide new structural insights into trans-dimerization of β1 and β2 subunits of the Na+,K+-ATPasePLOS ONE

Dear Dr. Shoshani,

Thank you for submitting your manuscript to PLOS ONE. After careful consideration, we feel that it has merit but does not fully meet PLOS ONE’s publication criteria as it currently stands. Therefore, we invite you to submit a revised version of the manuscript that addresses the points raised during the review process. Specifically, both reviewers commented that the manuscript contains a few grammar errors that should be addressed. Concern over the lack of description of the cluster analysis used in the study as well as what glycan residues are included. Please adderess the points raised by both reviewers in a revised manuscript.

We look forward to receiving your revised manuscript.

Kind regards,

Colin Johnson, Ph.D.

Academic Editor

PLOS ONE

“CONAHCYT (Proyecto Ciencia Frontera CF-2023-G-1454).”

“MMA thanks CONAHCYT for the Ciencia Frontera Project

CF-2023-G-1454.

MMA and GRS thank Computing Cluster Xiuhcoatl-Cinvestav (granted by

LANCAD) and Hector Manuel Oliver Hernandez.

JLRT thanks SIP-IPN 20240801, SIBE-IPN and EDI-IPN.”

“CONAHCYT (Proyecto Ciencia Frontera CF-2023-G-1454).”

5. Please provide a complete Data Availability Statement in the submission form, ensuring you include all necessary access information or a reason for why you are unable to make your data freely accessible. If your research concerns only data provided within your submission, please write "All data are in the manuscript and/or supporting information files" as your Data Availability Statement.

6. We notice that your supplementary figures and tables are included in the manuscript file. Please remove them and upload them with the file type 'Supporting Information'. Please ensure that each Supporting Information file has a legend listed in the manuscript after the references list.

Reviewers' comments:

Reviewer's Responses to Questions

**Comments to the Author**

1. Is the manuscript technically sound, and do the data support the conclusions?

Reviewer #1: Yes

Reviewer #2: Yes

2. Has the statistical analysis been performed appropriately and rigorously? 

Reviewer #1: I Don't Know

Reviewer #2: Yes

3. Have the authors made all data underlying the findings in their manuscript fully available?

Reviewer #1: No

Reviewer #2: Yes

4. Is the manuscript presented in an intelligible fashion and written in standard English?

Reviewer #1: Yes

Reviewer #2: No

5. Review Comments to the Author

Reviewer #1: This is an interesting paper using computational protein structure analyses to explore the differences in intercellular interactions of two homologs of Na,K-ATPase beta subunit. The authors are clear in their assumptions and in the use of a number of modeling methods to predict the extent and stability of the different homotypic protein-protein interactions. The conclusions are compared to what is known about the ability of the proteins to anchor cells into clusters.

There are minor questions to be answered.

1. Fig. 1 What is the orientation relative to alpha subunit. I.e. are we looking at it from the side or the top?

2. Which exact amino acid residues are included in the models?

3. Fig. 1, 2, and 8. It would be welcome to have stereo images in a supplement.

4. Please make it clear what glycan residues are included in the model. It seems to be only GlcNAc on the asparagines; is this right? Can you present an estimate of the minimum volume occupied by the actual complex glycan present in vivo, and whether the volume is enough to block interaction. Reference 45 from Schachner implicates oligomannosidic groups.

5. I could not understand Fig. 8 because the two models have different relative tilts. Can you add a duplicate of the figure with the position of alpha subunit L7-8 and position of beta transmembrane span for b1-b1, and the predicted positions of L7-8 and the transmembrane span for b2-b2? I think that L7-8 ought to dock into each model. If not, that should be stated, and I think that the model could be rebuilt including L7-8. Isn’t the interaction with alpha a constraint on the beta head models?

6. P. 31. What was achieved by producing a model of b1 with the Swiss Model program? How did it differ from 3WGU?

English:

eventhoh should be even though, and remove the comma

This should be one sentence with a comma: “The normalized energy per residue is around -2 kJ/mol in various conformers of β1-β1. While in β2-β2 none of the conformers reached that value.”

Molecular dynamics simulations along mathematical applications are... Along with?

Reviewer #2: The paper presents a well-executed study using MD simulations to investigate the differences in stability between the β1 and β2 subunits of the Na+,K+-ATPase in the context of their potential to form trans-dimers. The focus is on analyzing the structural stability in trans-dimerization of these subunits, with additional discussion on the role of glycosylation. However, it is important to note that no comparison is made with 3D structures lacking glycosylation, which limits the depth of the glycosylation analysis.

The manuscript would benefit from a language revision, as several sections contain unclear sentences. The use of references could be improved. In some sections, reference numbers are used instead of author names in the middle of a sentence, which disrupts the flow of the text.

Due to the lack of an available crystal structure for ATP1B2 the homologous pig gastric H,K-ATPase was used to build the model for the β2 subunit. For the analysis of the 3D models there is a discrepancy in the reporting of Ramachandran plots. For β1, the text mentions that 99% of residues are in allowed regions, while β2 is reported as 100%. However, the legend for Figure 1 claims none of the residues are in disallowed regions, yet the plot indicates that some residues are indeed in disallowed zones. This needs to be clarified.

In Table 2, the title incorrectly refers to the β1 dimer, while the content clearly discusses the β2 dimer. This should be corrected.

One important point that needs to be addressed is the lack of description of the cluster analysis which is used to draw conclusions about the stability of β1 and β2 dimers. The cluster analysis is central to the paper’s conclusions, yet there is insufficient explanation of the methods and presentation of the results. Supplementary Figure S2 suggests potential alternative interpretations, which should be discussed more thoroughly. Figure S2 should be combined with figure 7 for this discussion.

Finally, the concluding remark speculating on β2's interaction with a neural receptor to gain stability in trans-dimers may be somewhat premature. While the idea is intriguing, it may be more productive to explore potential interactions between β2 and other components of the Na+,K+-ATPase complex, e.g. the γ/FXYD subunits.

6. PLOS authors have the option to publish the peer review history of their article (what does this mean?). If published, this will include your full peer review and any attached files.

Reviewer #1: No

Reviewer #2: **Yes: **Hjalmar Brismar

---

## [Author Response · Author response to Decision Letter 1]

22 Dec 2024

Reviewer #1: This is an interesting paper using computational protein structure analyses to explore the differences in intercellular interactions of two homologs of Na,K-ATPase beta subunit. The authors are clear in their assumptions and in the use of a number of modeling methods to predict the extent and stability of the different homotypic protein-protein interactions. The conclusions are compared to what is known about the ability of the proteins to anchor cells into clusters.

There are minor questions to be answered.

1. Fig. 1 What is the orientation relative to alpha subunit. I.e. are we looking at it from the side or the top?

Thanks. We modified the orientation of the two models in Fig 1. Now We added this information to the figure legend.

“The extracellular domain of the β1 and β2 subunits depicted in A and C is with the N-terminal (residues 63 and 70) colored in yellow and the C-terminal (residues 303 and 289) in green. The distinctive alpha-helix of the β-subunits is localized in the bottom. For relative orientation to the alpha subunit, see S7 Figure.”

2. Which exact amino acid residues are included in the models?

ATP1B1 model shows the following range of residues: 63 to 303 for both of the chains.

ATP1B2 model shows the following range of residues: 70 to 289 for both of the chains.

This information was added to the text. Thanks.

3. Fig. 1, 2, and 8. It would be welcome to have stereo images in a supplement.

The suggested stereo images are now added as S3-S6 figures .

4. Please make it clear what glycan residues are included in the model. It seems to be only GlcNAc on the asparagines; is this right?

Thanks for the observation. As stated in “Participation of N-glycosylated residues in the dimeric interface” section; the type of glycosylations that were considered for the proteic models was N-acetylglucosamine (GlcNAc) to the nitrogen atom of an asparagine (N) side chain. We were also interested in including more sugars and evaluated the possibility to glycosylate in silico the beta-subunits of our interest. CHARMM-GUI has implemented Glycan Reader which permits the simulation of glycans and glycoconjugates. Glycan Reader not only detects most sugar types and chemical modifications in the PDB, but also allows users to edit the glycan sequences through addition/deletion/change of sugar types, chemical modifications, glycosidic linkages, and anomeric states. However, CHARMM-GUI Glycan Reader does not support in silico glycosylation and addition of a sugar at the reducing end of an existing glycan chain. (https://academic.oup.com/glycob/article/29/4/320/5301306). Therefore, we had to leave it as it is, with the GlcNAc molecule only.

Can you present an estimate of the minimum volume occupied by the actual complex glycan present in vivo, and whether the volume is enough to block interaction. Reference 45 from Schachner implicates oligomannosidic groups.

About ref. 45 (first version) this work was realized with mouse brain membranes. We can not be certain that human b2 has the same composition.

Anyway, this is indeed an interesting point for discussion. Unfortunately, we could not find any computational tool for calculating the sugar volume. Nevertheless, the steric effect of the sugars on the protein surface was considered in different studies and in fact, there are studies showing that it is relevant for both stabilizing the protein structure and for de-stabilizing it. Nevertheless, the N-glycosylation effect on protein-protein interactions is much more complex than simply analyzing its steric impediments. Different studies with a biophysical approach, analyzed the role of the N-linked sugar on protein folding and protein aggregation. Thus, Gavrilov et al. (2015; DOI: 10.1021/acs.jpclett.5b01588) showed that although natural glycosylation results in protein stabilization; in vitro and in silico studies show that sometimes glycosylation results in thermodynamic destabilization. Though glycosylation creates new short-range glycan–protein interactions that stabilize the conjugated protein, it breaks long-range protein–protein interactions. The destabilization originates not from simple loss of interactions but due to a trade-off between the short- and long-range interactions. Another interesting work published recently (Doran-Romana et al. 2024; DOI: 10.1126/sciadv.adk8173) shows the role of N-glycosylation as a protective mechanism against protein aggregation in eukaryotic cells. Nevertheless, as the focus of our study is on protein-protein interaction, we looked for studies that analyze this aspect. Qasba (2000) reviewed studies about the involvement of sugars in protein–protein interactions. This review emphasizes the complexity of that issue. “The oligosaccharide moiety in the carbohydrate-dependent recognition process orients the molecules in a way that brings about specific protein–protein or protein–carbohydrate interactions. As these interactions occur with a unique conformer of the oligosaccharide, the knowledge of the conformation of carbohydrates is important. A given oligosaccharide can exist in several conformations (Rao, Qasba, Balaji & Chandrasekaran, 1998), and it is the interaction between a unique conformer and a macromolecule that is required to initiate a biological response. Hence, it is essential to have detailed information about all the conformers that are accessible to an oligosaccharide.”

In the case of β2/AMOG, it is assumed to participate in both protein-protein (Roldan et al. 2022) and protein carbohydrate interactions (Heller et al. 2003). This makes it even more complicated. We do not know the carbohydrate structure nor its conformers, as the N-linked oligosaccharides of the human β2/AMOG were not studied or reported, at least we did not find that information. There is no crystal for β2/AMOG of any species and if it was, probably it would not have information about the linked sugars, as they would interfere with the crystallization of the protein.

All that discussion is out of the main scope of our work and requires much more studies therefore we do not include all of it in the manuscript but do make some clarifications along the manuscript that are highlighted in there.

5. I could not understand Fig. 8 because the two models have different relative tilts. Can you add a duplicate of the figure with the position of alpha subunit L7-8 and position of beta transmembrane span for b1-b1, and the predicted positions of L7-8 and the transmembrane span for b2-b2? I think that L7-8 ought to dock into each model. If not, that should be stated, and I think that the model could be rebuilt including L7-8.

The requested figure has been added as Fig. S7. The addition of elements into the system would of course modify the dynamics of the dimers, but we believe our results highlight a dramatic contrast in the tendencies of these dimers.

Isn’t the interaction with alpha a constraint on the beta head models?

Yes, of Course, and therefore, one of the criteria for selecting the most probable dimer (for b1-b1 and b2-b2) out of the docking results, is the orientation of the beta ectodomain. Selecting those that represent a trans orientation on membranes of two neighboring cells as mentioned in the manuscript. On the other hand, the domain that interacts with L7-8 of the alpha subunit is not involved in beta-beta interface. The adhesive interface is in the c-terminal (Ig-like) domain of the beta-subunit. At least for the beta1 subunit we have shown that the soluble extracellular domain (secreted from transfected CHO cells) is adhesive and mutations in hot spot residues of the proposed interface decreases cell-cell and protein-protein adhesion (Paez et al. 2019).

6. P. 31. What was achieved by producing a model of b1 with the Swiss Model program? How did it differ from 3WGU?

The identity between Human b1 subunit (P05026) and wild boar b1 subunit (3WGU) is 92.41%. As we were interested in studying the intermolecular interactions, we had to consider that the 7.6% difference in sequence could result in a different behaviour of protein-protein interactions. Therefore it was important to work with a structural model. When we compare the 3D structure of those two proteins, we find a homology of 95.5%.

Thanks, we have made this correction.

This should be one sentence with a comma: “The normalized energy per residue is around -2 kJ/mol in various conformers of β1-β1. While in β2-β2 none of the conformers reached that value.”

Ok, this paragraph has been corrected.

Molecular dynamics simulations along mathematical applications are... Along with?

Ok, this paragraph has been corrected.

Here is our reply (in blue) to Reviewer #2.

Reviewer #2: The paper presents a well-executed study using MD simulations to investigate the differences in stability between the β1 and β2 subunits of the Na+,K+-ATPase in the context of their potential to form trans-dimers. The focus is on analyzing the structural stability in trans-dimerization of these subunits, with additional discussion on the role of glycosylation. However, it is important to note that no comparison is made with 3D structures lacking glycosylation, which limits the depth of the glycosylation analysis.

1) The manuscript would benefit from a language revision, as several sections contain unclear sentences.

The use of references could be improved. In some sections, reference numbers are used instead of author names in the middle of a sentence, which disrupts the flow of the text.

Thanks for the observation. New references were added:

We found one case for the number instead of the author name and corrected it.

2) Due to the lack of an available crystal structure for ATP1B2 the homologous pig gastric H,K-ATPase was used to build the model for the β2 subunit. For the analysis of the 3D models there is a discrepancy in the reporting of Ramachandran plots. For β1, the text mentions that 99% of residues are in allowed regions, while β2 is reported as 100%. However, the legend for Figure 1 claims none of the residues are in disallowed regions, yet the plot indicates that some residues are indeed in disallowed zones. This needs to be clarified.

Thanks for the observation. We corrected it in the figure legend. It now says: “Ramachandran plot of the β1 subunit monomer where it can be seen that only 1% of the residues of the proteins are included in the disallowed regions”.

3) In Table 2, the title incorrectly refers to the β1 dimer, while the content clearly discusses the β2 dimer. This should be corrected.

Thanks. The title of Table 2 has been corrected.

4) One important point that needs to be addressed is the lack of description of the cluster analysis which is used to draw conclusions about the stability of β1 and β2 dimers. The cluster analysis is central to the paper’s conclusions, yet there is insufficient explanation of the methods and presentation of the results.

OK. We added an explanation in a separate section in “Methods” titled: PCA-based cluster analysis.

5) Supplementary Figure S2 suggests potential alternative interpretations, which should be discussed more thoroughly. Figure S2 should be combined with figure 7 for this discussion.

The discussion of the regions with the highest density has been rewritten. We hope it is clearer now.

6) Finally, the concluding remark speculating on β2's interaction with a neural receptor to gain stability in trans-dimers may be somewhat premature.

You are right. This idea at the concluding remark was misunderstood. We now comment on the heterophilic interaction of the b2-subunit in “Discussion” and hope it is now more coherent and not speculative.

While the idea is intriguing, it may be more productive to explore potential interactions between β2 and other components of the Na+,K+-ATPase complex, e.g. the γ/FXYD subunits.

It is indeed interesting and we added this possibility in the discussion. Nevertheless, our interest and focus is on the trans-interaction with another beta-subunit or another unknown protein on the neuron surface.

---

## [Decision Letter · Decision Letter 1]

8 Jan 2025

PONE-D-24-40078R1In silico studies provide new structural insights into trans-dimerization of β1 and β2 subunits of the Na+,K+-ATPasePLOS ONE

Dear Dr. Shoshani,

Thank you for submitting your manuscript to PLOS ONE. After careful consideration, we feel that it has merit but does not fully meet PLOS ONE’s publication criteria as it currently stands. Therefore, we invite you to submit a revised version of the manuscript that addresses the points raised during the review process. Specifically the reviewers require the correction of technical errors in the manuscript including problems with the representation of symbols and greek letters in the document file. In addition it was also thought that enlargement of the stereo images in the supplemental figure would be helpful to the reader.

We look forward to receiving your revised manuscript.

Kind regards,

Colin Johnson, Ph.D.

Academic Editor

PLOS ONE

Journal Requirements:

Reviewers' comments:

Reviewer's Responses to Questions

**Comments to the Author**

1. If the authors have adequately addressed your comments raised in a previous round of review and you feel that this manuscript is now acceptable for publication, you may indicate that here to bypass the “Comments to the Author” section, enter your conflict of interest statement in the “Confidential to Editor” section, and submit your "Accept" recommendation.

Reviewer #1: (No Response)

Reviewer #2: (No Response)

2. Is the manuscript technically sound, and do the data support the conclusions?

Reviewer #1: Yes

Reviewer #2: Yes

3. Has the statistical analysis been performed appropriately and rigorously? 

Reviewer #1: Yes

Reviewer #2: Yes

4. Have the authors made all data underlying the findings in their manuscript fully available?

Reviewer #1: Yes

Reviewer #2: Yes

5. Is the manuscript presented in an intelligible fashion and written in standard English?

Reviewer #1: Yes

Reviewer #2: No

6. Review Comments to the Author

Reviewer #1: Just a very minor request: for the stereo pictures in the supplement, they could be view at larger magnification if there was less white space between the images.

Reviewer #2: While the manuscript appears to be scientifically sound and likely acceptable for publication, the formatting issues in the submitted version are concerning. It reflects poorly on the submission process when such apparent errors are overlooked. Specifically, Greek letters are represented as empty boxes in the PDF, which makes it impossible to perform a thorough final review and provide a definitive recommendation in its current state.

Before resubmitting, I strongly urge the authors to double check that the manuscript is fully readable and that all formatting issues are resolved. See already on first page with several formatting errors.

Also correct Line 285: use the greek letter beta for the subunits as in other parts of the manuscript.

7. PLOS authors have the option to publish the peer review history of their article (what does this mean?). If published, this will include your full peer review and any attached files.

Reviewer #1: No

Reviewer #2: **Yes: **Hjalmar Brismar

---

## [Author Response · Author response to Decision Letter 2]

5 Feb 2025

Estimated reviewers

We appreciate the time and efforts that you have dedicated to read and revise our manuscript again.

Here is our reply (in blue) to Reviewer #1.

Reviewer #1: Just a very minor request: for the stereo pictures in the supplement, they could be view at larger magnification if there was less white space between the images.

The images (Fig S3-S6) were corrected and they are now larger. Thanks.

Here is our reply (in blue) to Reviewer #2.

1. While the manuscript appears to be scientifically sound and likely acceptable for publication, the formatting issues in the submitted version are concerning. It reflects poorly on the submission process when such apparent errors are overlooked. Specifically, Greek letters are represented as empty boxes in the PDF, which makes it impossible to perform a thorough final review and provide a definitive recommendation in its current state.

We are very sorry for that. As you could notice by downloading the Word Document itself, not the PDF, the symbols and Greek letters are OK. This problem was generated by the system, when converting all uploaded files to one PDF document.

The corresponding author revised the document before submitting it, actually the system does not permit to continue if the author does not open the file and revise it. For a strange reason, we did not notice this error and we apologize for it.

2. Before resubmitting, I strongly urge the authors to double check that the manuscript is fully readable and that all formatting issues are resolved. See already on first page with several formatting errors.

-It is double checked now and we hope you will get it as it is.

3. Also correct Line 285: use the Greek letter beta for the subunits as in other parts of the manuscript.

Thanks for your observation. It is now in Greek letter.

---

## [Decision Letter · Decision Letter 2]

3 Mar 2025

In silico studies provide new structural insights into trans-dimerization of β1 and β2 subunits of the Na+,K+-ATPase

PONE-D-24-40078R2

Dear Dr. Shoshani,

We’re pleased to inform you that your manuscript has been judged scientifically suitable for publication and will be formally accepted for publication once it meets all outstanding technical requirements.

Kind regards,

Colin Johnson, Ph.D.

Academic Editor

PLOS ONE

Additional Editor Comments (optional):

Reviewers' comments:

Reviewer's Responses to Questions

**Comments to the Author**

1. If the authors have adequately addressed your comments raised in a previous round of review and you feel that this manuscript is now acceptable for publication, you may indicate that here to bypass the “Comments to the Author” section, enter your conflict of interest statement in the “Confidential to Editor” section, and submit your "Accept" recommendation.

Reviewer #1: (No Response)

Reviewer #2: All comments have been addressed

2. Is the manuscript technically sound, and do the data support the conclusions?

Reviewer #1: (No Response)

Reviewer #2: Yes

3. Has the statistical analysis been performed appropriately and rigorously? 

Reviewer #1: (No Response)

Reviewer #2: Yes

4. Have the authors made all data underlying the findings in their manuscript fully available?

Reviewer #1: (No Response)

Reviewer #2: Yes

5. Is the manuscript presented in an intelligible fashion and written in standard English?

Reviewer #1: (No Response)

Reviewer #2: Yes

6. Review Comments to the Author

Reviewer #1: (No Response)

Reviewer #2: Thanks, all comments regarding formatting of greek letters have been fully adressed. I look forward to the published version of this interesting paper.

7. PLOS authors have the option to publish the peer review history of their article (what does this mean?). If published, this will include your full peer review and any attached files.

Reviewer #1: No

Reviewer #2: **Yes: **Hjalmar Brismar

---

## [Editor Report · Acceptance letter]

PONE-D-24-40078R2

PLOS ONE

Dear Dr. Shoshani,

I'm pleased to inform you that your manuscript has been deemed suitable for publication in PLOS ONE. Congratulations! Your manuscript is now being handed over to our production team.

Kind regards,

on behalf of

Dr. Colin Johnson

Academic Editor

PLOS ONE